# Atmospheric Density and Temperature Vertical Profile Retrieval for Flight-Tests with a Rayleigh Lidar On-Board the French Advanced Test Range Ship *Monge*

**Robin Wing** [1,2,*] **, Milena Martic** [1,2]**, Alain Hauchecorne** [1,2]**, Jacques Porteneuve** [1,2]**,**
**Philippe Keckhut** [1]**, Yann Courcoux** [3]**, Laurent Yung** [4]**, Patrick Retailleau** [4] **and Dorothee Cocuron** [5]

[1] Laboratoire Atmosphères, Milieux, Observations Spatiales (LATMOS), Institut Pierre Simon Laplace (IPSL), UVSQ Université Paris-Saclay, Sorbonne Université, CNRS, 78280 Guyancourt, France; Milena.Martic@latmos.ipsl.fr (M.M.); Alain.Hauchecorne@latmos.ipsl.fr (A.H.); Jacques.Porteneuve@latmos.ipsl.fr (J.P.); Philippe.Keckhut@latmos.ipsl.fr (P.K.)

[2] Gordien Strato, 11 Boulevard d'Alembert, 78280 Guyancourt, France

[3] Direction Générale de l'Armement DGA, 60 bd Gén Martial Valin, 75015 Paris, France; yann.courcoux@intradef.gouv.fr

[4] Direction Générale pour l'Armement, Direction des Essais, DGA-Essais de Missiles, BEM Monge, BCRM BREST, C.C. 51, F-29240 BREST CEDEX 9, France; laurent.yung@intradef.gouv.fr (L.Y.); patrick.retailleau@intradef.gouv.fr (P.R.)

[5] Direction Générale pour l'Armement, Essais de Missiles site Landes, 40115 Biscarrosse Air, France; dorothee.cocuron@intradef.gouv.fr

**\*** Correspondence: robin.wing@latmos.ipsl.fr

**Abstract:** The Advanced Test Range Ship Monge (ATRSM) is dedicated to in-flight measurements during the re-entry phase of ballistic missiles test flights. Atmospheric density measurements from 15 to 110 km are provided using one of the world's largest Rayleigh lidars. This lidar is the culmination of three decades of French research experience in lidar technologies, developed within the framework of the global Network for Detection of Atmospheric and Climate Changes (NDACC), and opens opportunities for high resolution Rayleigh lidar studies above 90 km. The military objective of the ATRSM project is to provide near real time estimates of the atmospheric relative density profile, with an error budget of less than 10% at 90 km altitude, given a temporal integration of 15 min and a vertical resolution of 500 m. To achieve this aim we have developed a unique lidar system which exploits six laser transmitters and a constellation of eight receiving telescopes which maximises the lidar power-aperture product. This system includes a mix of standard commercially available optical components and electronics as well as some innovative technical solutions. We have provided a detailed assessment of some of the more unique aspects of the ATRSM lidar.

**Keywords:** Rayleigh lidar; middle atmosphere; temperature; density; gravity waves

## 1. Introduction

The Advanced Test Range Ship Monge (ATRSM) seen in Figure 1 is the main French facility which enables in-flight measurements of ballistic missile tests during the re-entry phase [1]. Given that the re-entry phase of a missile trajectory spans the entire atmospheric column at a speed of hundreds of meters per second and that the ship-board measurements can occur in very rough ocean conditions, obtaining high quality and reliable measurements of mesospheric density and temperature perturbations is a technically challenging feat. In this article we lay out the scientific challenges of making measurements of Upper Mesospheric and Lower Thermospheric (UMLT) atmospheric density

and temperature above 90 km using a single measurement technique, describe a unique ultra high powered Rayleigh lidar experiment which can achieve the required temporal and vertical resolution for the ATRSM mission objectives, present some selected results of the ATRSM lidar, and discuss several possible collaborative research projects with atmospheric scientists.

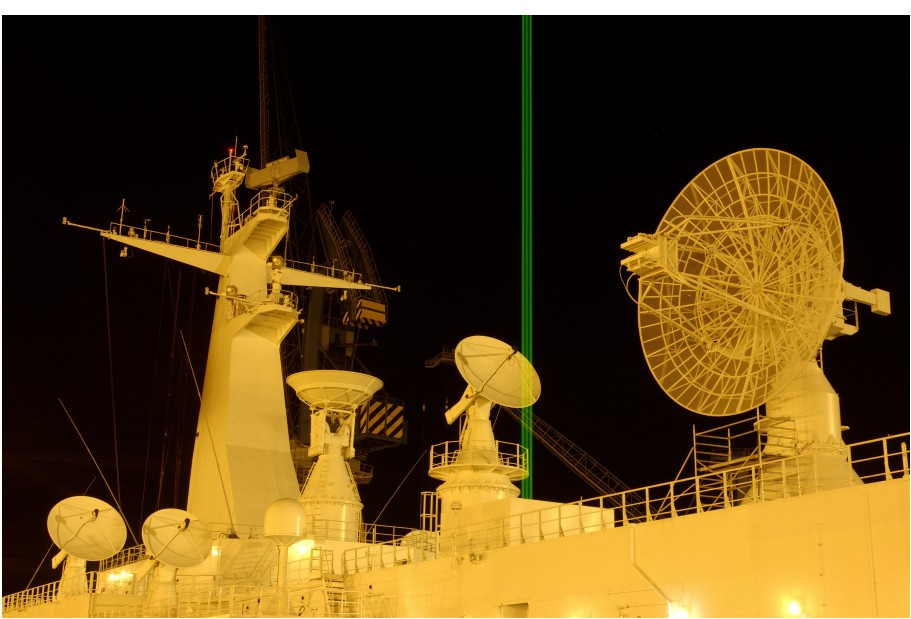

**Figure 1.** Triple laser beam seen exiting the deck of the Advanced Test Range Ship Monge.

Initially, UMLT measurements have been performed using in-situ techniques like rocketsondes, falling spheres or rocket-borne balloon sondes [2], however these methods often have large uncertainties due to aerodynamic and thermal corrections [3]. For example, when a falling sphere payload passes the supersonic falling speed regime (Mach = 1) around 70 km, the measured temperature has a warm biases of nearly 5 K with respect to temperature profiles from remote sensors [4].

Other measurement techniques used for long-term temperature monitoring in the UMLT, such as rocketsondes, satellites, airglow measurements, falling spheres, and indirect imaging techniques, have been reviewed by [5]. A key conclusion of this report highlights the increasing difficulty and cost of launching rockets for meteorological and aerological measurements and the promising development of high power Rayleigh lidar sensors which exploit elastic backscatter form an active transmitter to directly infer the atmospheric density. A complete description of the technique is given in Sections 1.1 and 1.2. In addition to Rayleigh lidars, major contributions for UMLT studies have been made from metal resonant fluorescence lidars, such as Na lidar [6], Fe lidar [7] and potassium lidar [8].

A new Rayleigh lidar, based on the measurement principles developed at the end of the seventies [9], was designed for the French ATRSM to measure high-altitude temperature and density profiles during ballistic missile test flights. The lidar is intended to collect data at UMLT altitudes where the missile components are submitted to high physical stress during re-entry phases. Given the short duration of the flight and the even smaller window associated with the critical re-entry phase in which we seek to make measurements, the main goal of this lidar project is to produce an estimate of the atmospheric state at the highest possible altitude in the shortest integration time, with a maximum reliability. The lidar measurements are conducted before, during and after a missile test flight in order to provide the best possible reconstruction of the atmospheric conditions along the missile re-entry pathway given reasonably small geographic and temporal separation.

The ATRSM lidar is the one of a series of French Rayleigh lidars operating in the stratosphere and mesosphere [10]. This system was developed and implemented using the experiences garnered from the installation and operation of three ground observatories and one ship-borne lidar. French

lidar studies have been occurring since 1978 at the Observatoire de Haute Provence (OHP), the Centre d'Essais des Landes (CEL), the Observatoire de Réunion (OR), and on-board the Advanced Range Test Ship Henri Poincaré (ARTSP). Data provided by some of these long running lidar experiments are considered as international references for stratospheric and mesospheric temperature trends under the auspices of organisations such as the Network for the Detection of Atmospheric Composition Change (NDACC) and other climate monitoring and satellite validation organisations [11–13].

The feasibility of stratospheric and mesospheric Rayleigh lidar measurements at sea without requiring adaptive optics in the primary receiving telescope was demonstrated in the 1980s, by both Russian [14] and French [10] research groups. Ship-borne lidars are particularly valuable nowadays for atmospheric investigations given the dearth of sounding profiles over the ocean [15]. The ATRSM lidar was commissioned in 1992, was retrofitted in 2005, and has been receiving regular hardware and software upgrades since (see Table 1) The lidar is operated in the same configuration as it's predecessor the ATRSP aboard the Henri Poincaré and is primarily a campaign focused instrument. lidar measurements are routinely compared to ship-launched radiosondes measurements of temperature and density between 0 and 30 km as well as to satellite data and atmospheric models at higher altitudes.

In this publication, Sections 1.1 and 1.2 contain a summary of the lidar method, Section 1.3 describes the main objectives of the lidar on-board the ATRSM, Section 2 details the unique design and configuration of the ATRSM lidar including some innovative designs, Section 3 reports on the lidar density observations, capabilities and uncertainties, Section 4 shows a recent case-study of gravity wave activity which highlights the utility of the ATRSM system, finally in Section 5 some conclusions and future prospects are provided.

### 1.1. Temperature, Pressure and Density Methodology

Lidar systems transmit laser pulses, when a laser pulse is sent vertically into the atmosphere some small fraction of the photons back-scatter off atmospheric molecules and particles. The returned photons can be collected and digitised using a telescope receiver array, optical filters, and photon counting devices. The shape of the photon counts profile is proportional to the atmospheric density profile and can be normalised using a model (e.g., MSIS [16]). An example of this procedure can be found in Barton et al. [17]. Corrections for changes in molecular composition above 90 km, arising from the differential molecular weight of various major atmospheric constituents ($N_2$, $O_2$, O, and Ar), are calculated to produce a more accurate Rayleigh density profile [18]. Corrections for count rate saturation can be made by estimating the deadtime and applying a correction factor [19]. By splitting the photon acquisition system into a high gain and low gain channel and re-combining the independent corrected signals we can further extend the dynamic range of our measurements.

Calculating pressure, $P(z)$, from our relative density profile, $\rho(z)$, requires the assumption of hydrostatic equilibrium Equation (1) and the Ideal Gas Law Equation (2), where $g(z)$ is gravitational acceleration as a function of altitude, $R$ is the gas constant, and $M$ is the molecular mass. By assuming an a priori 'seed pressure' (e.g., MSIS) at the top of the lidar profile we can iterate Equation (3) downwards in altitude to generate a pressure profile. A poor choice of a priori will distort the top portion of the pressure profile as the seed pressure pulls the measurement towards the model climatology. To partially compensate for this effect we calculate the pressure profile multiple times, reducing the altitude where we take the a priori pressure with each iteration, until the average of the first 10 km in the lidar pressure pressure is within 20% of the average model pressure for the same altitude range.

$$dP(z) = -\rho(z)g(z)dz \tag{1}$$

$$P(z) = \frac{R\rho(z)T(z)}{M} \tag{2}$$

$$\frac{dP(z)}{P(z)} = -\frac{Mg(z)}{RT(z)}dz = d(\log(P(z))) \tag{3}$$

By following the technique outlined by [9] we can integrate the expression for pressure Equation (1) to derive the absolute temperature profile, $T(z)$, Equation (5).

$$X_i = \frac{\rho(z_i)g(z_i)\Delta z}{P(z_i) + \frac{\Delta z}{2}} \tag{4}$$

$$T(z_i) = \frac{Mg(z_i)}{R\log(1 + X_i)}\Delta z \tag{5}$$

This technique is very efficient throughout the middle atmosphere and can be used to make high vertical resolution profiles of temperature, pressure, and density.

### 1.2. Temperature, Pressure and Density Uncertainty

The primary contribution to the uncertainty budget in Rayleigh lidar measurements is statistical in nature and can be described by using Poisson counting statistics for detecting back scattered photons. The standard error for a Poisson distribution can be estimated by the square root of the number of counted photons. Likewise the statistical error for a lidar density or temperature profile can be obtained by propagating the standard error of the photon counts through Equations (1) to (5). The relative error of photon counts, density and temperature is approximated in Equation (6) where $\Delta\rho(z)$ is the relative uncertainty in the density profile, $\Delta T(z)$ is the relative uncertainty in the temperature profile, $\Delta N(z)$ is the relative uncertainty in the photon counts profile, and $\beta$ is the background count rate due to electronic noise and ambient light. The relative contribution of the statistical error at a given altitude can be reduced by integrating the lidar photon counts profile in both altitude and time. The integration acts to increase the number of photons in each counting bin by reducing the resolution of the lidar measurement.

$$\frac{\Delta\rho(z)}{\rho(z)} \approx \frac{\Delta T(z)}{T(z)} \approx \frac{\sqrt{N(z) + \beta}}{N(z)} \tag{6}$$

The a priori uncertainty (also called the tie-on error or seed error), which arises from the aforementioned assumption of model pressure at the top of the lidar profile as seen in Equation (1), is also a significant contributor in the uncertainty budget. Previous work has been done attempting to model the contributions of the a priori estimate to the resulting pressure and temperature profiles [20] and the resulting recommendation is to remove the top 16 km of the lidar temperature profile to avoid contamination. This approach is not ideal for the objectives of the Monge project which seeks to evaluate the atmospheric conditions above 90 km at high temporal resolution. The new Optimal Estimation Technique (OEM) [21–23] for Rayleigh lidar retrievals has resolved this issue and fully describes the uncertainty budget. The technique, while powerful and effective, does not yet have an NDACC validated standardised algorithm available for public use. We look forward in future work to developing this technique for use abroad the Monge with the ATRSM lidar.

As was stated in Section 1.1 we use an iterative approach to generate the lidar pressure profile and recalculate the profile until we converge on a more stable solution. The cut off threshold for our iteration is 20% relative uncertainty at the top of the pressure profile.

The largest uncertainty terms in the ATRSM lidar error budget in the UMLT are the statistical Poisson error and the smoothing error which arises from integrating the lidar profile in altitude. The uncertainty budget in the lower stratosphere below 25 km is dominated by the presence of aerosols. If high concentrations of aerosols are present the lidar will suffer a pronounced cold bias with respect to co-located radiosondes. A comparison of ATRSM lidar profiles in a relatively aerosol-free atmosphere is shown in Section 3.2.

### 1.3. Measurement Goals and Requirements

Atmospheric entry is the process by which vehicles (e.g., ballistic missile) which are outside the atmosphere can penetrate that atmosphere and reach the surface in good condition. The reentry vehicle accelerates under gravitational forces until the first perceptible aerodynamic effects occur between 120 and 90 km [24]. Ballistic missiles are only guided during the powered phase of flight and during the unpowered phases of early reentry they have simple ballistic trajectories which can be altered by atmospheric perturbations. The challenges of ballistic reentry are to reduce the aero-thermic drag heating, to prevent the full vaporisation of the payload, to deliver a payload to a predetermined target, and to show the smallest atmospheric signature to avoid the early detection of the warhead. To meet these challenges the Monge lidar must be able to furnish high resolution measurements of density, pressure, and temperature the upper mesosphere in order to study the aerodynamic effects of small-scale atmospheric perturbations on flight trajectories.

The complete reentry phase of a ballistic trajectory in Earth's atmosphere occurs at altitudes below 120 km, however the critical phase of the reentry process corresponds to the peak heat flux that occurs in the mesosphere around 60–80 km. This phase can be easily influenced by large variations of environmental conditions and as a consequence of the atmospheric friction and drag can lead to incredible external heating (several 1000 K) of the missile and surrounding air. The mean free-body forces acting on the missile in this region are the mean aerodynamic forces, namely the gravitational force and the thrust force. The aerodynamic perturbations to the mean free-body state during orbital reentry and atmospheric descent are generally described by drag deceleration lift coefficients which are a function of the atmospheric density and temperature perturbations. Using a model of the missile shape and dimensions, as well as the density, pressure, and temperature perturbations measured by the lidar we can model the flow conditions and calculate the Mach number, Reynolds number and Knudsen number to control for surface deflections, aerodynamic angles, angular descent rates.

It is well understood that the source of these body-force perturbations are short period (less than an hour) internal gravity waves which can greatly influence on the momentum budget of the mesosphere [25]. These waves propagate upwards in the atmosphere and modify the local density, pressure, temperature and wind fields. The quantification of small scale gravity wave-driven temperature changes with lidar have been subject of many publications in the 1990s [26–29] and lidar climatologies of gravity waves have been developed by long running NDACC lidar stations [30,31]. One of the key conclusions of these studies is that density anomalies have widely varying amplitudes from day-to-day and site-to-site due to regional and local meteorological conditions caused by convection, ageostrophy and orographic effects. These density and temperature anomalies are spread over a wide spectrum of wavelengths and are not easy to predict at a given location and time based on a climatology. In the extreme case we can see that wave-driven mesospheric inversions layers can change local density by up to 30% with respect to climatological models.

For these reasons, the best way to provide realistic near real time estimates of density and temperature profiles along a vehicle reentry trajectory over the north Atlantic ocean is to use near-real time Rayleigh lidar remote sensing on-board a ship.

## 2. Instrument Description

### 2.1. Optics and Lasers

The ATRSM lidar system can be categorised as one of the largest Ultra Large lidars (ULL) in the world. A rough metric used to quantify a lidar's measurement capability is the Power-Apeture Product (PAP) which is simply the output laser power of the transmitter multiplied by the collector surface area. The Monge lidar approaches this problem by employing an array of eight 0.5 m receiver telescopes and six 24 W lasers for a total PAP of up to 226.1 W m$^2$ depending on the number of lasers used. A 3-D schematic of the ATRSM lidar can be seen in Figure 2.

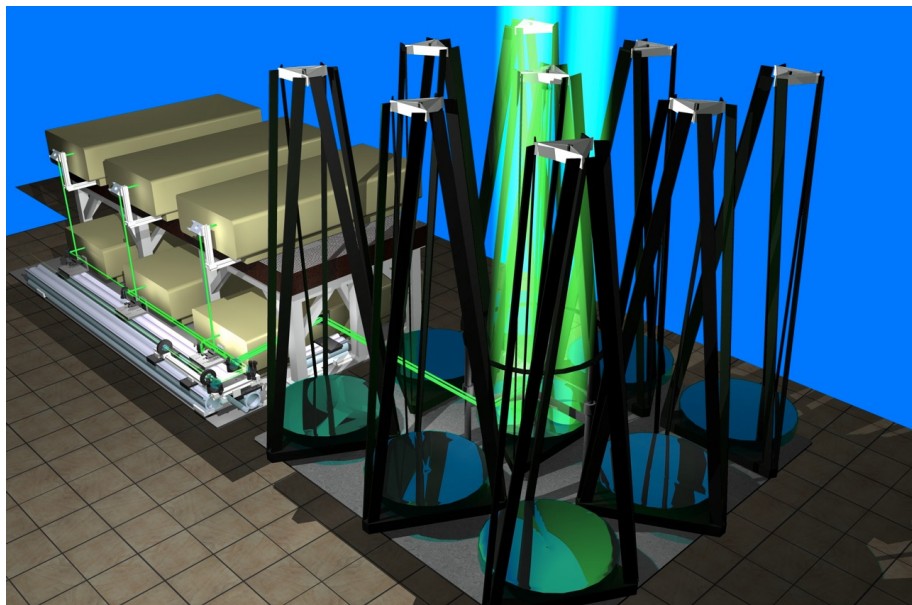

**Figure 2.** Configuration of the ATRSM lidar showing six laser transmitters, eight 50 cm receiver telescopes, and one 50 cm transmitting telescope.

A few other lidars in this class include: the Observatoire de Haute Provence in Southern France which has four 0.5 m telescopes and a 24 W (at 532 nm) laser (PAP = 19.2 W m$^2$) [10,13], the US Airforce 100 Inch lidar at Dayton which had a 2.54 m telescope and a 5.3 W at 532 nm laser (PAP = 26.9 W m$^2$) [32], the Maïdo lidar at La Réunion which has a 1.2 m telescope and two 12 W at 355 nm lasers (PAP = 54.3 W m$^2$) [33], The Purple Crow lidar with a unique 2.65 m liquid mercury telescope and a 30 W at 532 nm laser (PAP = 165.6 W m$^2$) [34], the Alomar Observatory lidar in Norway with two 1.8 m telescopes and two 20 W (divided between 355 nm and 532 nm) lasers (PAP = 203.2 W m$^2$) [35] and the Utah State University (USU) lidar with four 1.25 m telescopes two 21 W at 532 nm lasers (PAP = 206.6 W m$^2$) [36].

There are two general strategies to maximise the PAP: (1) either have a large area for collecting photons or (2) have a high laser power. The Alomar and Purple Crow lidars are excellent examples of maximising the area of the receiving telescopes. The ATRSM lidar system, while exhibiting a large telescope constellations (seen in Figure 2), also exploits the second approach and combines the output of six 24 W Nd:YAG lasers for an average laser output of 144 W. The two largest ULL systems, the ATRSM and USU lidar systems, exploit both strategies to increase the PAP.

Combining and aligning six outgoing laser beams is a unique technical challenge. In order to accommodate all the light in the outgoing optical path two techniques were used for coupling the lasers. First, the laser beams are coupled two-by-two using polarisation cubes to create three laser beams (see Figure 3).

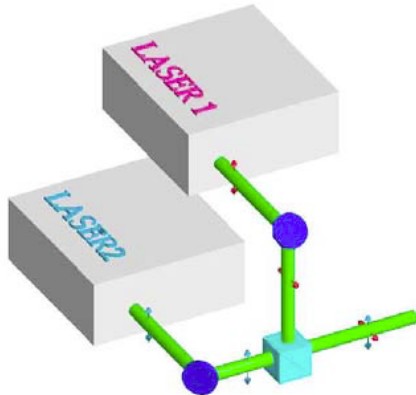

**Figure 3.** Schema showing the recombination principle of two lasers by polariser cube.

Second, each of the three beams is then transmitted through a set of 532 nm coated mirrors mounted on the transmitting telescope where each beam (10 mm diameter) is expanded to 15 cm diameter beam and reflected off the 50 cm parabolic transmitting telescope (see Figure 4). The transmitting telescope directs the laser light outside the lidar room through an optical window which is mounted to the ship's roof. This window protects all the optical components in a rough maritime environment enabling measurements even during rough seas or blowing sea-spray. The divergence of the three outgoing laser beams is better than 0.5 mrad and the parallax distance between the telescope axes and the emitter is between 67 and 95 cm depending on the mirror. Full recovery of the geometric overlap function is obtained by 12.5 km.

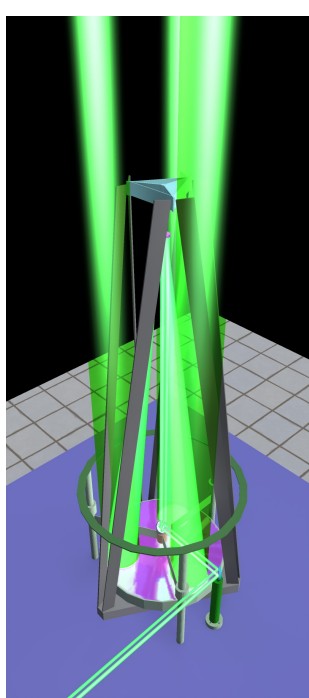

**Figure 4.** Optical configuration showing the emission of 3 beams using a single telescope.

## 2.2. Electronics and Signal Acquisition

The back-scattered photons which are collected in the 8 mirror telescope constellation are counted using new, compact Hamamatsu R9880U-210 mini-photomultipliers (Hamamatsu Photonics, Hamamatsu City, Japan). The mini-PMTs are cooled to reduce thermal noise in the dynode chain which is important for correctly estimating the true 'sky-noise' in the signal background. The signal

gating is set to 15 and 30 km for the high and low gain Rayleigh channels (respectively) and is required to avoid signal induced noise on the PMTs due to initial signal returns generated at low altitudes.

Photon counting is done using a Licel PR10-160 (Licel, Berlin, Germany) digital/analog transient recorder with a resolution of 16384 channels of 15 m each. The combination of the analog and digital channels in the Licel allows for a greater dynamical range in the counting system.

**Table 1.** Technical modifications made to the lidar system onboard the Monge.

|  | 1992–2005 | 2005–2019 |
|---|---|---|
| laser energy (mJ/pulse) | 800 | 4000–5000 @ 30 Hz |
| laser repetition rate (Hz) | 30 | 30, 60, 90, 180 |
| polarisation cubes | 0 | 3 |
| telescope area (m$^2$) | 1.57 | 1.57 |
| emission divergence (μrad) | 50 | 33 |
| field of view (mrad) | 0.3 | 0.2 |
| telescope parallax (cm) | 67–95 | 67–95 |
| photomultipliers | R928 (water cooling) | R928/R9880U-20-TEC |
| Filter FWHM (nm) | 1 | 0.3 |
| Transient Recorder | SESO | Licel PR 20-160-P/PR10-160P |

## 3. Signal Characterisation

To maximise the efficacy of the ATRSM lidar profile at the highest possible temporal resolution we require fast photon counting and low Signal to Noise Ratio (SNR) at high altitudes. For this study SNR is defined as the count rate at 40 or 60 km divided by the count rate above 120 km which is a combination of the sky background, and electronic and thermal noise in the detector chain. The SNR at 60 km for selected measurements is shown in Figure 5. The red trace shows the SNR of the high gain channel which is optimised for single photon counting in the UMLT and the blue trace shows the SNR for the low gain channel which is used to correct the high gain channel below 40 km. Both curves are characterised by a slow decline from 2013 to 2017 due to degradation of the telescope optics due to marine conditions and ageing electronics. In 2018 both channels were upgraded with new PMTs and counting electronics (see Table 1) and the resulting jump in SNR can bee seen midway across Figure 5. All nine of the 50 cm telescopes are set to be replaced by the end of 2019 and we expect another similar jump in the ATRSM SNR curves to appear in future version of this figure.

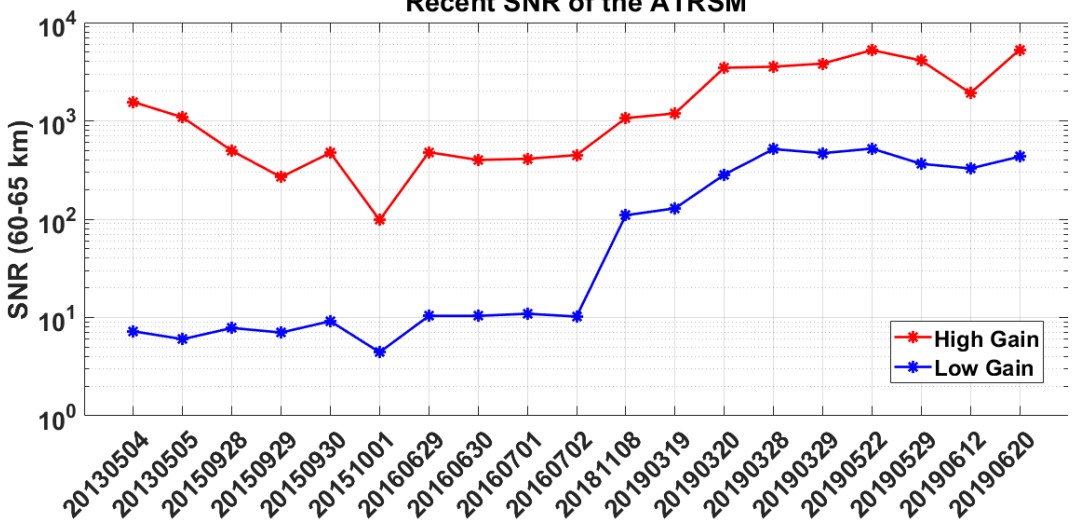

**Figure 5.** ATRSM lidar SNR for selected nights 2013–2019. High gain (red) and low gain (blue) channels both show improvement after the hardware upgrades of 2018.

### 3.1. Density Characterisation

Recall that the ATRSM mission objective is to provide near real time estimates of the atmospheric relative density profile with an error budget of less than 10% at 90 km given a temporal integration of 15 min and an effective vertical resolution of 500 m. Presented in Figure 6a are four consecutive lidar density profiles, calculated at 500 m vertical integration and normalised using the MSIS model (black reference line). We can immediately see the benefit of the real-time lidar measurement over the climatological model. MSIS does not accurately portray the instantaneous density scale height below 65 km which is evidenced by the slope in the normalised density. As well the lidar measurement shows a strong, quickly evolving, wave-driven density perturbation in a layer above 70 km.

In Figure 6b we show the associated relative uncertainty associated with the 15 min density profiles shown in Figure 6a. We can see that the uncertainty increases exponentially with altitude and has an uncertainty of 7 % at 90 km.

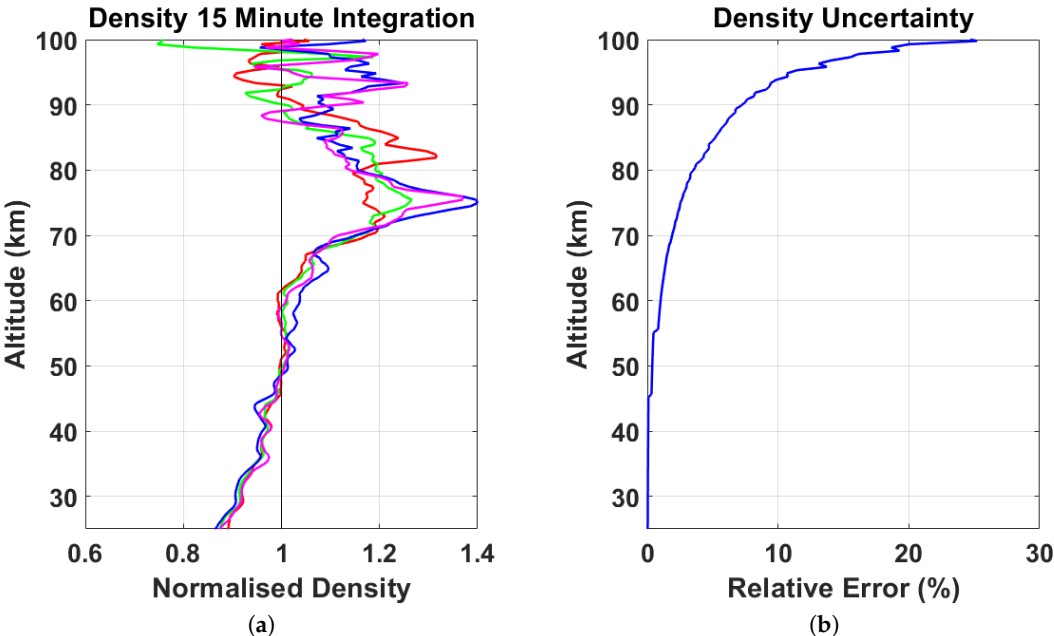

**Figure 6.** Characterisation of ATRSM normalised density profiles at 500 m vertical resolution and 15 min integration. (**a**) Four normalised density profiles produced using 15 min integration of ATRSM data prior to a missile re-entry; (**b**) The relative uncertainty for a 15 min integration density profile from ATRSM.

### 3.2. Temperature Characterisation

Temperature profiles produced from the ATRSM lidar are routinely compared to temperature profiles from radiosondes which are launched from the ship while at sea. The agreement between the lidar and the radiosondes, in the region of measurement overlap between 25 and 33 km, is generally very good. Figure 7 shows a series of 15 min integration lidar temperature profiles starting from half an hour after the radiosonde launch. There is generally good agreement between the lidar the radiosonde temperatures. We can see several small scale structures in the radiosonde temperature profile which are also represented in the lidar profile. The slight 1–3 K temperature offset below 26 km and the larger 2–8 K offset above 28 km are likely due to either sampling of different air masses or a cold bias resulting from aerosol contamination. This particular balloon flight was sampling air 36 km away from the ship at 25 km altitude, 51 km away at 30 km, and 66 km away at 33 km. The perennial problem of coincidence between remote sensing techniques and in-situ measurements can provide additional sources of uncertainty for the estimation of local aerodynamic coefficients associated with

missile reentry. To positively identify this 1–3 K cold bias as aerosol contamination a vibrational $N_2$ Raman channel is required.

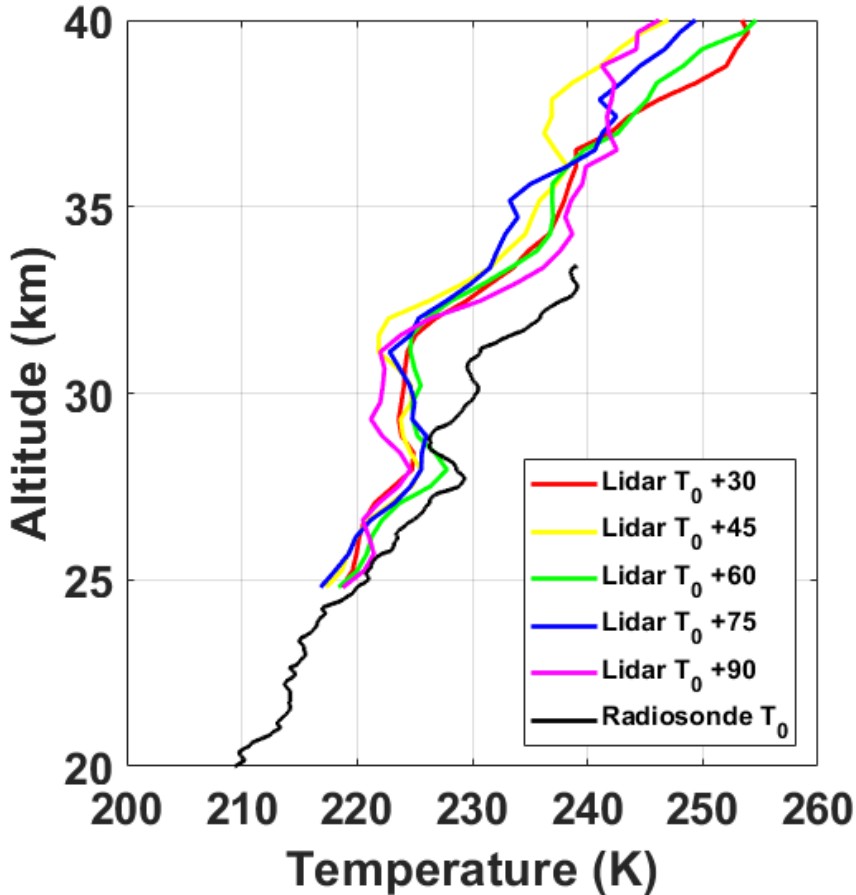

**Figure 7.** Five 15 min integration lidar temperature profiles calculated after the launch of a radiosonde from onboard the ATRSM. Lidar profiles are shown 30, 45, 60, 75, and 90 min after radiosonde launch from on board the Monge.

Figure 8 shows the benefit of combining the high gain and low gain elastic Rayleigh channels in the ATRSM lidar. Here we present three temperature profiles which are calculated from the low gain photon counts channel (blue), the high gain photon counts channel (red) and a combination of the two channels (green). For clarity we have added red dots on the melded green profile to signify areas where the melded profile is identical to the profile produced using only the high gain channel. The melding of the high and low gain channels is accomplished by a simple uncertainty-weighted average of the scaled photoncount profiles (based on an adaptation of Equation (6)). From Figure 8 we can see that the temperature profile produced from the melded photon counts profile (green) is not significantly influenced by the low gain channel above 55 km and is identical to the temperature profile produced from the high gain channel. Below 45 km the melded temperature profile is similar to the temperature profile produced from the low gain channel and does not suffer from the same warm bias as the high gain channel which results from counting saturation in the PMT. In the region between 45 and 55 km the melded temperature is generally between the temperatures from the two independent channels.

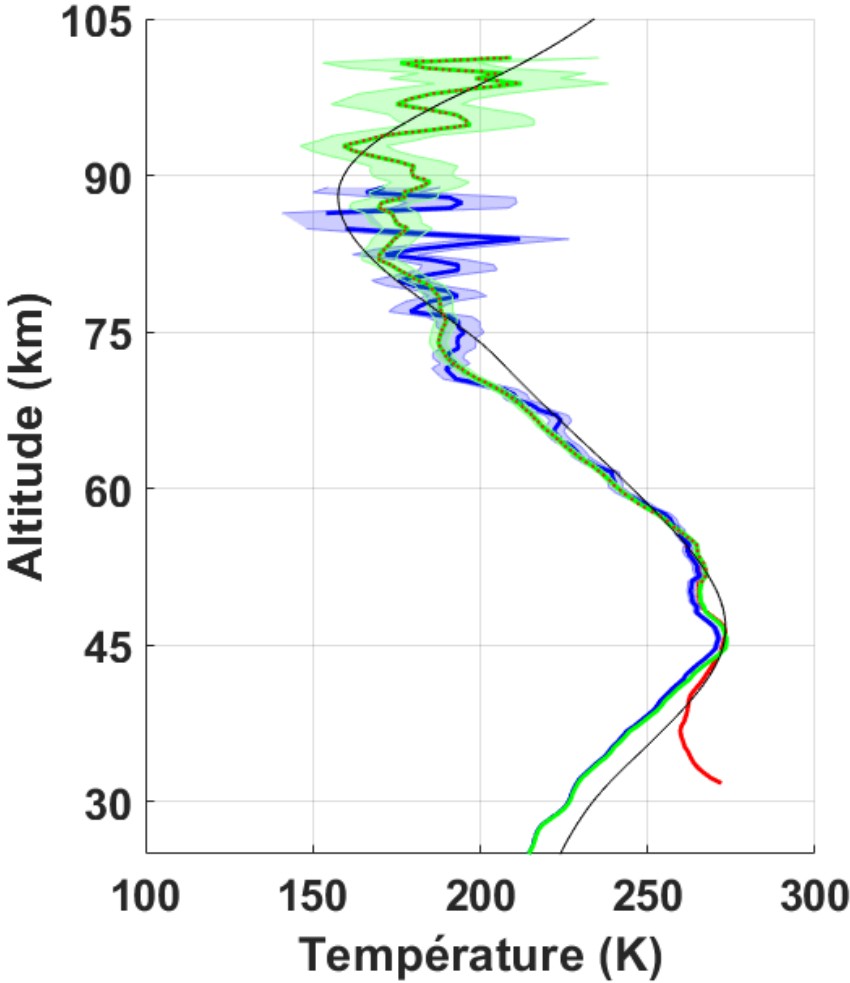

**Figure 8.** A one hour temperature profile calculated using the high gain channel (red), low gain channel (blue), and combined lidar signal (green). Red dots represent where the melded profile and the high gain only profile are identical. Black line is the temperature from the MSIS model.

## 4. Gravity Wave Case Study

Presented in this section is a case study of a rapidly evolving and breaking short period gravity wave observed from the ATRSM in summer 2019 over the North Atlantic. In the upper right panel of Figure 9a we see a temperature profile calculated from one hour of ATRSM lidar measurements. We can see some evidence of wave activity near 75 km where the temperature lapse rate becomes inverted in the mesosphere. Figure 9b gives the relative error for the temperature profile as a function of altitude and establishes that the inversion layer at 75 km is significant.

The lower two panels Figure 9c,d show the same 60 min of ATRSM lidar data integrated at 30 min and 15 min respectively. The relative uncertainty at 30 min and 15 min is not meaningfully different from the profile presented in Figure 9b. For clarity, the shaded uncertainty is not added to the bottom two panels.

In the 30 min integration in Figure 9c we see that the gravity wave develops in the first half hour of the measurement (blue) and begins to strongly break and force the mesospheric inversion layer in the second half hour (red). In the 15 min integration in Figure 9d we see more clearly the dynamics of the gravity wave. The first temperature at $T_0$ (red) shows a distinct double inversion layer with peaks at roughly 71 km and 84 km. Fifteen minutes later in the profile (green) we see the lower peak of the wave has moved up to 75 km and has begun to break. The higher peak has also begun to diminish

as the energy of the wave is dissipated at 75 km. Half an hour after $T_0$ we see the inversion at 75 km reaches its maximum and the temperature lapse rate above the inversion layer relaxes back towards the adiabatic lapse rate. The last 15 min (magenta) of the measurement we see signs that the amplitude of the inversion is beginning to decrease however, this is not significant at this level of integration. Additionally, we do see a significant oscillation in the temperature of the stratopause.

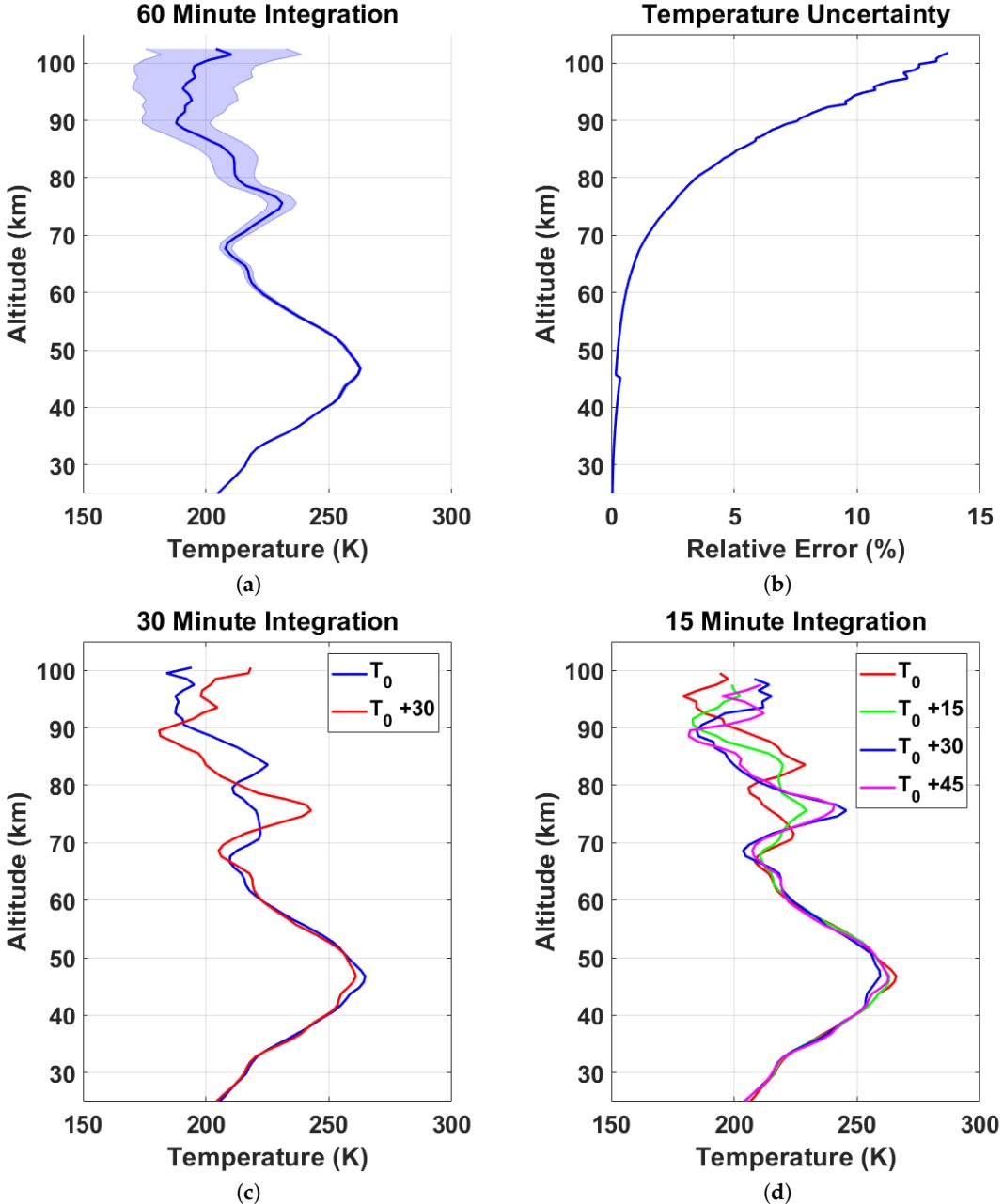

**Figure 9.** Characterisation of ATRSM temperature profiles at 500 m vertical resolution and 15 min integration. (**a**) A temperature profile produced using a 60 min integration of ATRSM data prior to a missile re-entry; (**b**) The relative uncertainty for a 60 min integration temperature profile from ATRSM. Differences in the uncertainty profile due to 30 and 15 min integration are negligible; (**c**) Two temperature profiles produced using a 30 min integration of ATRSM data prior to a missile re-entry. Wave activity is clearly seen above 75 km; (**d**) Four temperature profiles produced using a 15 min integration of ATRSM data prior to a missile re-entry. The evolution of a cresting gravity wave is clearly seen above 75 km.

## 5. Conclusions

The requirements for the ATRSM project specify that the lidar can furnish near real time profiles of density and temperature perturbations in the UMLT. We have demonstrated that for a typical cloud-free night we can produce a density profile at 500 m vertical resolution and 15 min integration which has a relative uncertainty of less than 10% at 90 km (the uncertainty was 7% in the case presented).

In the context of the case study presented in Section 4 we can see that the state of the atmosphere can be extremely variable near 80 km, where aerodynamic effects become important, on time scales similar to those of the missile re-entry. By furnishing near real-time profiles of density and temperature perturbations we can meet the measurement needs of the technicians responsible for calculating ballistic trajectory parameters.

The ATRSM lidar provides an excellent development platform for conducting further studies of short time scale, high resolution studies of UMLT dynamics. In particular, further studies of gravity wave breaking and turbulence could be undertaken.

**Author Contributions:** Conceptualization, R.W., M.M., A.H. and Y.C.; methodology, R.W., M.M., J.P. and A.H.; software, R.W.; validation, A.H.; formal analysis, R.W.; investigation, R.W.; resources, J.P.; data curation, M.M., Y.C., L.Y. and P.R.; writing–original draft preparation, R.W., Y.C., and M.M.; writing–review and editing, M.M.; visualization, J.P. and R.W.; supervision, M.M., P.K., and D.C.; project administration, M.M.; funding acquisition, M.M. and D.C. All authors have read and agreed to the published version of the manuscript

**Funding:** This research was funded by Direction Générale pour l'Armement Français and Gordien Strato.

**Acknowledgments:** The first prototype onboard the Henri Poincaré was developed by SESO, while the upgraded lidar was provided by Gordien Strato Sarl, we thank A. Colpin, G. Cerruti, A. Garnier, C. Laqui, B. Legac, F. Keckhut, A. Gaillado, B. Stein (Licel), and O. Lang for their contributions.

**Conflicts of Interest:** The authors declare no conflict of interest.

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
