# Peer review of "Atmospheric Density and Temperature Vertical Profile Retrieval for Flight-Tests with a Rayleigh Lidar On-Board the French Advanced Test Range Ship Monge"

_atmosphere, doi:10.3390/atmos11010075_

Round 1
Reviewer 1 Report
This paper details impressive improvements to an existing remote sensing instrumental technique and presents an interesting application for the use of the new instrument’s measurements. Overall, the paper is well-written and logically organized. More detailed analysis could improve the paper’s claim that it is capable of meeting all of the requirements for the proposed application.
Major comments:
Lines 79-87: How are you modifying the Rayleigh lidar temperature reduction (Hauchecorne and Chanin, 1980) in order to account for changing composition with diffusive equilibrium occurring above around 90 km? Above 90-100 km, there is more atomic oxygen created by photodissociation and the atmosphere is no longer well-mixed.
Lines 139-143, Section 4, 285-287: I’m not completely convinced of your statement in Lines 285-287 from what you’ve shown/described in the paper. Please clarify how the Rayleigh data is intended to be used to support the ballistic missile tests/modeling/detection. Is the purpose of this lidar to characterize small perturbations in the atmosphere to influence atmospheric re-entry models? You mention measuring near-real time thermodynamic profiles, is this to be used as a tactical aid? If so, how can you assure that the narrow lidar beam will overlap with the missile’s trajectory? It would be good to provide some more discussion (in a Discussion section or in the Conclusion section) on how this lidar’s measurement capabilities meet the requirements for this application. For example, are the lowest time scales of 15 minutes acceptable for the modelling needs mentioned in Lines 139-143? I think you could cover this in a paragraph or less.
Note that some of the comments that follow are grammatical and stylistic suggestions, so changes to not need to be made for every comment.
Minor comments:
Lines 21-26: These sentences might serve better at the end of the Introduction (around line 64)
Line 27: The Rayleigh lidar technique has also been around for decades, so it has also been “historically” used. Perhaps “Initially” or another word would work better here.
Line 32: Please give examples of the “other techniques”.
Line 35: It would be helpful to add a sentence briefly describing the Rayleigh lidar technique and mentioning it will be discussed in more detail later.
Line 36: “A specific Rayleigh…” try “A new Rayleigh...”
Line 35-36: Be consistent with lidar vs. LIDAR capitalization
Lines 39-43: Are the Rayleigh lidar measurements meant to characterize the background state of the atmosphere for this application or actually detect a change in the atmosphere as a missile passes by? If it’s the latter, how will you be able to solve the needle-in-the-haystack issue of having the missile trajectory pass right through the lidar’s field of view? If possible, please give more detail on how this instruments measurements will be used in relation to missile tests.
Line 55: Need to distinguish between Rayleigh and Mie scatter.
Line 61: Is there a reference for previous versions of the system you can give?
Line 62: Are the radiosondes launched from the ship. Please clarify.
Line 71: Need to note you’re not describing a lidar system: start sentence with something like: “Lidar systems transmit laser pulses, when a laser pulse…”
Line 75: This technique has been shown in:
Barton, D. L., et al. (2016), Variations in Mesospheric Neutral Densities from Rayleigh Lidar Observations at Utah State University. DOI: 10.1051/epjconf/201611913006
Line 75: Need to spell out and add reference for MSIS, try:
Picone, J. M., et al. (2002), NRLMSISE‐00 empirical model of the atmosphere: Statistical comparisons and scientific issues, Journal of Geophysical Research. DOI: https://doi.org/10.1029/2002JA009430
Line 81: Need to mention that molecular mass is only constant up to about 90 km and address how you will or justify why you will not modify the Rayleigh temperature algorithm.
Equations 1-3: These equations could be integrated better into the text in the paragraph above
Lines 91-92: The Rayleigh lidar technique mathematically described above is not used in aerosol lidar, metal-resonance lidar, or water vapor lidar.
Line 98: Why haven’t you shown or discussed error propagation of the Poisson raw lidar data uncertainty through the temperature algorithm? Can you give some rough numbers for delta_rho/rho and delta_T/T to back-up the approximate relationship given in Eqn 6?
Line 108: I believe there is a more up-to-date version of this technique that’s been published:
Rica, R. J., and A. Haefele (2015), Retrieval of temperature from a multiple-channel Rayleigh-scatter lidar using an optimal estimation method, Applied Optics. DOI: https://doi.org/10.1364/AO.54.001872
Jalali, A., et al. (2018), Improvements to a long-term Rayleigh-scatter lidar temperature climatology by using an optimal estimation method, Atmospheric Measurement Techniques. DOI: https://doi.org/10.5194/amt-11-6043-2018
Jalali, A., et al. (2019), A practical information-centered technique to remove a priori information from lidar optimal-estimation-method retrievals, Atmospheric Measurement Techniques. DOI: https://doi.org/10.5194/amt-12-3943-2019
Line 108: I also believe that the method described in the references above allows you to retain more of the high altitude portion of the temperature profile.
Line 140-143: Can you provide a reference to modeled data or your own figure to show the flow conditions around the missile? Or is this future work? Please clarify.
Line 156: I don’t think you’ve definitely proven in the text above that Rayleigh lidar is the only technique, could say it’s the best technique available today for this application, so something similar.
Line 164: State that it’s an array of 8 receiver telescopes.
Line 176-180: Really, the ATRSM and USU lidars combine both strategies and maximize laser power and telescope area. Please clarify.
Line 185-193: I would call it a “transmitting telescope” instead of an “emission telescope”.
Line 206: Please clarify what you mean “noise” in your definition of signal to noise ratio (SNR). Is it purely the Poisson noise in the measurement, purely the sky background, or a combo of the two? What about electronic noise terms like thermal noise, dark counts, etc.?
Lines 216-224 and Figure 6: Please define, mathematically what you mean by “normalized”. I would think this just means scaling the lidar densities by an MSIS number density, but the units on the horizontal axis are not in #molecules/m^3. Did you also calculate a difference?
Line 221: Please clarify what “instantaneous scale height” means.
Lines 228-241 and Figure 7: Nice plot. I would expect aerosols to have less of an effect as you go up in altitude and it looks like you have larger disagreement between the lidar and radiosonde at higher altitudes. To me, this suggests that the effect of the increasing horizontal distance between the two and sampling different air masses has the largest effect on the two techniques agreement. Please add a note in the caption that the additional numbers in the legend are in minutes.
Figure 8: Another nice figure. Please state what the black line represents (MSIS temperatures?). Also, please add a legend, space permitting.
Figure 9: In Figure 9d it’s visually hard to discern the time evolution of the wave activity. Could you instead stack the profiles so they are separated from one another horizontally and restate the horizontal axis as “relative temperatures”? Or could you could show as contour plot with time on the horizontal axis and temperature as the color scale.
Section 4 and Figure 9: Why did you choose to show temperatures as opposed to temperature perturbation plots (where the small-time scale profiles are compared to a nightly average). Please clarify your choice of gravity wave analysis.
Reviewer 2 Report
The paper describes a powerful lidar system on a floating platform to conduct upper mesosphere lower thermosphere temperature and atmospheric density measurements. The floating platform, ATRSM in this case, will inevitably cause vibration in the pointing angle of lidar beam, thus, generating the uncertainty in the lidar altitude ranging (converting LOS distance to altitude). I do not see the author address how this system corrects this uncertainty, or any estimation on the magnitude of this uncertainty, in the manuscript. Alos, when talking about UMLT lidar temperature observations, I think it is important to recognize the important contributions from metal resonant fluorescence lidar technology, such as Na lidar (Krueger et al., 2015), Fe lidar (Gelbwachs, 1994) and potassium lidar (von Zahn and Hoffner, 1996).
